# Development of a Universal Validation Protocol and an Open-Source Database for Multi-Contextual Facial Expression Recognition

**DOI:** 10.3390/s23208376

**Published:** 2023-10-10

**Authors:** Ludovica La Monica, Costanza Cenerini, Luca Vollero, Giorgio Pennazza, Marco Santonico, Flavio Keller

**Affiliations:** 1Department of Engineering, Unit of Computational Systems and Bioinformatics, Università Campus Bio-Medico di Roma, 00128 Rome, Italy; ludovica.lamonica@unicampus.it (L.L.M.); l.vollero@unicampus.it (L.V.); 2Department of Engineering, Unit of Electronics for Sensor Systems, Università Campus Bio-Medico di Roma, 00128 Rome, Italy; costanza.cenerini@unicampus.it; 3Department of Science and Technology for Sustainable Development and One Health, Unit of Electronics for Sensor Systems, Università Campus Bio-Medico di Roma, 00128 Rome, Italy; m.santonico@unicampus.it; 4Department of Medicine, Unit of Developmental Neuroscience, Università Campus Bio-Medico di Roma, 00128 Rome, Italy; f.keller@unicampus.it

**Keywords:** facial expression recognition, labelled data database, facial landmarks, machine-learning algorithms, affective computing

## Abstract

Facial expression recognition (FER) poses a complex challenge due to diverse factors such as facial morphology variations, lighting conditions, and cultural nuances in emotion representation. To address these hurdles, specific FER algorithms leverage advanced data analysis for inferring emotional states from facial expressions. In this study, we introduce a universal validation methodology assessing any FER algorithm’s performance through a web application where subjects respond to emotive images. We present the labelled data database, FeelPix, generated from facial landmark coordinates during FER algorithm validation. FeelPix is available to train and test generic FER algorithms, accurately identifying users’ facial expressions. A testing algorithm classifies emotions based on FeelPix data, ensuring its reliability. Designed as a computationally lightweight solution, it finds applications in online systems. Our contribution improves facial expression recognition, enabling the identification and interpretation of emotions associated with facial expressions, offering profound insights into individuals’ emotional reactions. This contribution has implications for healthcare, security, human-computer interaction, and entertainment.

## 1. Introduction

Emotionsare a fundamental form of nonverbal interaction among human beings, constituting two-thirds of all communication [1], and allowing for the interpretation of individuals’ intentions. Among the various channels of nonverbal communication, facial expression is one of the most important and informative in both interpersonal communication and human-computer interaction (HCI) [2,3]. It has been studied for many years, starting with Charles Darwin [4]—one of the first scientists to recognize the power of facial expression in detecting individuals’ emotions, intentions, and opinions.

### 1.1. Related Work

Facial expression recognition remains a complex challenge [5,6]: achieving a comprehensive understanding and prediction of emotional manifestations requires a meticulous integration of various contextual components, including gender, age, ethnicity, and culture, as well as diverse modalities of emotion stimulation such as audio, images, video, or text. Moreover, it is crucial to be able to assess these factors in real-world settings for an accurate prediction and creation of realistic emotional experiences. To face these challenges, specific algorithms have been developed that leverage advanced data analysis tools to obtain information on the emotional state of individuals based on their facial expressions, i.e., Facial Expression Recognition (FER) algorithms [7]. These algorithms typically exploit one of the two fundamental theories of emotions, namely Ekman’s discrete emotion theory [8] or Russell’s dimensional theory of emotions [9], or in some cases both, to achieve a comprehensive and accurate classification of facial expressions. In any case, regardless of the method employed, technologies for detecting emotions through facial expressions analysis follow a similar process that begins with face identification, continues with the extraction of fiducial points, known as landmarks, and finally uses their positions to deduce the emotion linked to the observed expression [10].

The Facial Action Coding System (FACS) [11], developed by Ekman and Friesen in 1978, is arguably the most well-known and widely used system for analyzing facial activity. It is a manual that provides a detailed linguistic description of every possible motor alteration of the face, codifying them in terms of the so-called Action Units (AU), thus facilitating the objective evaluation of facial activity. Each AU is described in detail, explaining the muscular activation, position, direction of movement, duration, and intensity of muscular action, and how these factors can influence the perception of facial expressions by an observer. The automation of FACS would make it a cornerstone tool in the field of behavioral sciences; however, currently, despite numerous attempts it remains an unresolved practice. The difficulty of such automation lies in the fact that the association of emotions with facial expressions through the use of the manual is extremely complex and requires the presence of a human component, such as a qualified expert, capable of recognizing the emotional components from the frames eventually extrapolated from an Action Unit decoding system.

The basic emotion classification provided by Ekman [8] is adopted to categorize the emotional content of facial expressions: the six basic emotion categories (i.e., happiness, sadness, surprise, fear, anger and disgust), the neutrality component and the unique description of each basic emotion in terms of facial expression are used. Recent research suggests that timing plays a key role in how facial expressions are interpreted, which is why researchers are focusing on algorithms that enable real-time monitoring of the dynamics of facial emotions [12].

There are a variety of pre-made options available for creating FER applications with ease. The Emotion API is one of the services supplied by the Microsoft Cognitive Service Pack [13]; this API enables the identification of faces and their expressions. The emotions detected are classified using the six basic Ekman emotions, as well as the absence of emotion, identified as neutrality. This API only allows for the processing of images or videos, which means that they can only be decoded after being captured; as a result, this application lacks real-time implementation.

The MIT Media Lab founded Affectiva in 2009; the Affectiva technology, also known as Affdex [14], identifies 21 facial expressions in people and remaps them to Ekman’s six fundamental emotions. Particularly, Affectiva uses Paul Ekman’s Facial Action Coding System to classify the detected facial expression by analysing the pixels of 21 zones. Affectiva is one of the easiest technologies to be integrated into a new project. The technology from Affectiva is totally open source and provides an SDK for Windows, Java, Objective-C, C++, Unity, and JavaScript. This aspect simplifies integration into other systems, although the system’s performance in recognizing and identifying inner emotions is not satisfactory [15,16].

The six basic emotions outlined by Ekman are used by the majority of emotion APIs to classify facial expressions. Eyes, brows, nose, and mouth are main of the facial characteristics that are taken into account by most emotion APIs [17].

Another valid option is the face-api.js library. This open-source JavaScript library is built on top of the tensorflow.js core and implements various convolutional neural networks (CNN) to perform face detection and recognition [18]. Face-api.js is able to identify faces and estimate a series of information about them, including:Face detection: it identifies the boundaries of one or more faces.Face landmark detection: it extracts the position and shape of eyebrows, eyes, nose, mouth, lips, and chin.Face recognition: it identifies individuals in an image.Facial expression detection: it determines the expression of a face based on Ekman’s discrete theory of emotions, including the additional neutral class alongside the six basic emotions.

To fulfill the different tasks, face-api.js offers five pre-trained models: MobilenetV1, TinyFaceDetector, FaceLandmarkModel, FaceLandmark68TinyNet, and FaceRecognitionModel.

The MobilenetV1 and TinyFaceDetector models enable face detection in images and video streams. They were trained on the WIDER FACE dataset, which comprises 32,203 images and 393,703 labeled faces, demonstrating a high degree of variability in scale, pose, and occlusion. Subsequently, the FaceLandmarkModel and FaceLandmark68TinyNet models facilitate the detection of 68 facial landmarks on the recognized faces from the previous models. These models were trained on approximately 35,000 labeled images with 68 landmarks. Lastly, the FaceRecognitionModel calculates facial features. To accomplish this, it was trained on over three million images.

The utilization of these models makes the library particularly flexible, user-friendly, and computationally lightweight. Additionally, its extensibility allows for its integration into real-time web applications.

In general, despite the different solutions available, emotion detection in realistic circumstances still presents difficulties due to significant intra-class variation and low inter-class variation, such as changes in facial position and small variations between expressions. Additionally, facial expression recognition is a continuously expanding research area that requires a constant availability of high-quality labeled data [19]. Datasets of labeled data play a pivotal role in the development and evaluation of FER algorithms [20]. In fact, these data are commonly employed by researchers to train and assess their developed algorithms [21]. In light of this, various datasets of images or video frames have been developed and made publicly available for the research community to use in the development and validation of FER algorithms [21,22]. Notably, there are several publicly accessible datasets for facial emotion recognition, such as AffectNet, CK+ (Extended Cohn-Kanade), and FER-2013.

AffectNet represents one of the widely adopted dataset for detecting facial emotions [23]. This dataset comprises approximately one million facial images obtained from the internet using emotion-related keywords in six languages. Out of these, around 457,000 images were manually annotated by experts to associate them with the seven discrete facial expressions and the intensity of valence and arousal. The CK+ dataset [24], an extension of the CK (Cohn-Kanade) dataset [25], encompasses both video sequences and static images. Each image is labeled with one of seven emotions, which include the six basic emotions and an additional category for contempt. All images and videos were captured against a constant background. The FER 2013 dataset is widely utilized for facial emotion recognition and comprises 35,887 grayscale images collected from the internet and labeled by crowdsourced workers [26]. Further datasets for facial emotion recognition encompass EmoReact, MMI, and RAF-DB [21,22]. The choice of dataset largely depends on the specific task and available resources. Large-scale datasets can enhance the performance of deep learning models, while smaller datasets may be more suitable for evaluating traditional machine learning models.

### 1.2. Objectives

From the analysis of the state-of-the-art it emerged that the evaluation and verification of FER algorithms are challenging yet critical. For this reason, the first goal of this work is to devise a universal validation methodology, applicable to any algorithm, aimed at evaluating the performance of Facial Expression Recognition (FER) algorithms. To do so, a web framework was designed, capable of incorporating different algorithms for their investigation. Within this framework, a specific algorithm was tested on healthy subjects, recording their expressions, and comparing the results with their declared emotional state. To elicit an emotional response, the framework presents the user with highly emotive images carefully selected from a specific database. Due to its wide use in various research projects and application domains [27,28] even if lacking a thorough accompanying documentation, in this project we chose to validate the FER algorithm implemented in the JavaScript library face-api.js. From reverse engineering the code, we observed that the face-api.js face expression recognition model utilizes depthwise separable convolutions and densely connected blocks in its architecture based on ResNet, a popular model for many image recognition tasks due to its ability to train deep networks using skip connections or shortcuts. The model training dataset included diverse images from public datasets and the internet.

From the analysis of the literature, the necessity for annotated databases to develop FER algorithms has become apparent. Presently, existing datasets mainly consist of images, requiring processing through Convolutional Neural Networks (CNNs) or preprocessing to extract facial landmark coordinates for use in machine learning classifiers. Another crucial aspect pertains to the labeling of these datasets, which has been carried out by developers rather than by the subjects themselves, the very individuals from whom the facial expressions were captured. This phenomenon introduces a significant gap in the data, as it fails to fully capture the genuine emotional responses of the test participants.

Hence, the secondary objective of this study is to collect the subjects’ facial expressions’ data during the testing phase, thereby crafting an accurate and comprehensive dataset of labeled information. This dataset will serve as a valuable resource for training versatile FER algorithms that can discern users’ facial expressions with utmost precision, leveraging facial landmarks as the foundation for analysis.

As a final objective, we have planned to develop a FER algorithm that, leveraging machine learning methodologies, allows for the verification of the reliability of the database constructed, and can be used as an effective methodology for the classification of facial expressions based on facial landmarks coordinates.

Therefore, this research aims to provide a significant contribution to the understanding and development of techniques for facial expression recognition. This work consists of the presentation of a validation method that allows for verifying the reliability of any FER algorithm implementable in web applications. Additionally, this study includes the creation and release of a labeled database, named FeelPix, which can be used to train and test any FER algorithm that recognizes facial expressions based on facial landmarks coordinates. Furthermore, the research involves the development of a computationally lightweight yet sufficiently performant algorithm, enabling the evaluation of the effectiveness of the released database by recognizing facial expressions based on the data included within it.

## 2. Materials and Methods

### 2.1. Experimental Protocol

**Experimental setup.** A single subject at a time was tested using a web application in an environment without external sources of lighting, in order to make the results comparable without influencing the user’s expressiveness. In order to ensure uniformity of results among all participants, the same computer was used, a Lenovo Essential V15-IIL notebook with the following characteristics:Intel Core i7-1065g7 processor;8 GB Ram;Hd 512 GB SSD;Display 15.6${}^{\prime \prime }$;Windows 11;0.3 MP front camera resolution.

Google Chrome and Web Server for Chrome applications were used to run the web application in full-screen mode after preloading it from a specific folder.

The language used for the instructions, the evaluation system, and to communicate with participants was Italian.

**Protocol.** Each participant was provided the appropriate instructions regarding the test procedure before its execution. In addition, the graphical interface informed users about the progress of their tests and subsequent activities. Participants were also asked to timely report any discomfort they felt so that the experiment could be discontinued. A pool of 70 images was identified prior to the experimentation within a specific dataset of highly emotive images, in order to derive the images to be presented to the subjects. Each participant was presented with a randomized subset of 35 images in order to display five images for each of the seven considered emotions (i.e., the six Ekman basic emotions and the neutral emotional status), in a non-consecutive manner. This allowed for variation in the set and order of images presented to each subject, ensuring the generalization of the results thus enhancing the internal validity of the study.

The decision to present participants with a subset of 35 images was grounded in a meticulous assessment of test timing. During the study’s design phase, a series of pilot tests were conducted to determine the average time a participant would take to complete the emotional test using varying quantities of images. It was revealed that with a selection of 35 images, we were able to maintain the total test duration within 10 min. We deemed it essential to prevent participants from experiencing boredom or discomfort due to the test’s length, as this situation could adversely affect their emotional expressiveness. Additionally, an excessively long test duration could have led to increased variability in emotional responses, making data interpretation more challenging.

Concurrently, specific guidelines were followed in presenting images to the participants: first of all, five photos were displayed in a non-consecutive order for each emotion (i.e., neutrality, happiness, anger, fear, sorrow, disgust, and surprise). Each image was shown in full-screen mode for 3 s, and the face-api.js library was used to identify emotions and facial cruxes (i.e., landmarks). After this step, the image was reduced and participants were able to see the emotion rating scale. During the decision-making phase, each participant used a seven-button rating scale to describe the emotional state aroused. Each button is labeled with the associated emotion, supported by the corresponding emoticon. In order to avoid affecting the results, the detection of emotions was switched off throughout this decision-making phase. Furthermore, no time restriction was set on the participants in order to give them as much freedom as possible in their choice of emotional state. In fact, each participant was given the choice to move to the next image by pressing a specific button only when he or she thought it was suitable. However, this button was designed in a way that it could only be used after choosing at least one emotional state. Additionally, the web application allowed users to pick multiple emotions for each image simultaneously and change their minds until they had moved on to the next image. Finally, a break time was implemented between each image, during which the subject was informed about the test’s progress.

The task was completed in 5–9 min without applying any time restriction.

**Participants.** A total of 31 healthy subjects (13 females and 18 males) aged between 20 and 69 years participated in the test phase; the age distribution of the participants can be observed in the graph presented in Figure 1.

The data collected for each participant were: first name, surname, date of birth, gender and whether or not they would wear glasses during the test. All subjects voluntarily participated in the study, and no form of compensation was provided. The study was approved by the local ethical committee (see the reference number in the appropriate section at the end of the paper).

### 2.2. Selection of Highly Emotive Images

The selection of appropriate stimuli to consistently induce emotional states played a pivotal role in the experimental protocol. Established stimulus sets are crucial data sources, providing researchers with the means to control and manipulate experimental conditions effectively. At present, numerous well-established, highly emotive stimulus sets are readily accessible. These stimulus sets, carefully selected to evoke emotions consistently, undergo a rigorous standardization process aligned with either the discrete emotion theory and/or the dimensional theory of emotions. As a result of these comprehensive standardization efforts, a robust array of stimulus sets has been crafted to harmonize with established theoretical frameworks in the field of emotion.

In this study, it was decided to provide individuals with static visual stimuli, i.e., images, as suggested by specialized literature [29,30]. We preferred this stimuli over dynamic stimuli such as video or audio tracks, since the aim was to capture instantaneous emotions rather than the temporal evolution of emotional states whereas the use of dynamic stimuli such as video or audio tracks may have introduced confounding factors related to the length and complexity of the stimuli, potentially masking or altering the immediate emotional responses being investigated. Finally, the static images allowed for fine control over the emotional content presented to the participants.

Three different image databases were taken into account [31]: the International Affective Picture System (IAPS), the Open Affective Standardized Image Set (OASIS), and Nencki Affective Picture System (NAPS).

The first one, the IAPS database, developed by Lang et al. [32], provides normative ratings of emotions (pleasure, arousal, dominance) for a series of color images that constitute a set of normative emotional stimuli for experimental investigations of emotion and attention [33]. In this database, each item is accompanied by a set of norms (mean and standard deviation) along three dimensions: arousal (physiological activation evoked by the image), valence (pleasantness and pleasure) and dominance (the degree of control of the emotional state by the subject). More recently, the IAPS has also been standardized according to the discrete emotion theory. The usage of this database is granted by its owner upon request, with the condition that it is solely used for research purposes. The authors of this work applied for access to the database, but the approval for access has not yet been given.

The OASIS [34], is an open-access online stimulus dataset containing 900 color images depicting a broad spectrum of themes, including humans, animals, objects, and scenes, together with normative ratings on two affective dimensions: valence (i.e., the degree of positive or negative affective response that the image evokes) and arousal (i.e., the intensity of the affective response that the image evokes). The OASIS images were collected from online sources and the ratings of valence and arousal, expressed according to the dimensional theory of emotions, were obtained through an online study. The main advantage of this database lies in its free use for research purposes.

The last database considered is the Nencki Affective Picture System (NAPS) [35], which today is the richest database of visual stimuli with extensive semantic information and a complete set of related normative evaluations. It consists of 1356 realistic photos that have been split into five categories: people, faces, animals, objects, and landscapes. Each image is characterized by a series of emotive ratings that were collected through a test phase on 204 subjects. In particular, the analysis was carried out using the Self-Assessment Manikin (SAM) [36], i.e., a non-verbal pictorial evaluation technique which produces results based on the dimensional theory of emotions. Specifically, using the SAM, each participant expressed the emotional state elicited by the image in terms of valence, arousal, and approach-avoidance. In addition to emotional information, the NAPS provides physical attributes about the images such as brightness, contrast, and entropy. The NAPS was followed by three expansions, including the NAPS Basic Emotions (NAPS BE) [37], which was used in this study.

We decided to use the NAPS BE database rather than the OASIS database, as the algorithm under investigation in the experimental phase yields results based on the discrete theory of emotions to assess the emotional state of the subjects.

This database is a subset of 510 NAPS images that provides classifications based on both the discrete emotion theory and the dimensional theory of emotions. In particular, the NAPS BE includes images that belong into the same five categories as the original database split into the following quantities: 98 animals, 161 faces, 49 landscapes, 102 objects, and 100 people. In order to provide the characterization in discrete terms of this subset of images, a test phase with 124 subjects was developed by the database authors.

In this study, a carefully selected subset of NAPS BE images was used to reduce the number of considered elements and enhance the effectiveness of the experimentation. A selection method was developed to ensure impartiality and prevent our biases from affecting the screening of the images. This method, implemented in Matlab R2022a, utilizes the information provided by the information table associated with the database. Using this approach, we were able to isolate 70 images and divide them into seven groups, each corresponding to one of the seven emotional states considered by the discrete emotion theory. In particular, because images frequently had several labels associated with them, it was not possible to select and categorize them using only discrete labels. For this reason, for each image we took into consideration the intensity values associated with each of Ekman’s six basic emotions, provided in range [1, 7], and valence and arousal values, provided in range [1, 9].

The valence and arousal values were subjected to analysis using the k-means clustering algorithm, aiming to identify emotional clusters for grouping the images within the database. The reason for choosing this algorithm is that it always converges and has a very light computational overhead. During the process, a number of clusters equal to three was set in order to obtain three distinct groups of images, each of which could be associated with one of the discrete macro-categories of emotions introduced by Ferré et al. and Kissler [38,39], namely in *positive*, *negative* and *neutral*. At the same time, another relevant finding was the link, identified by the same authors, between the labels of these macro-categories and mean valence values, such as 2, 5 and 7. In fact, based on this information, each of the three clusters was associated with the corresponding label according to the following logic:*Negative* label to the cluster of images with valence values predominantly in the range [1, 4];*Neutral* label to the cluster of images with valence values predominantly in the range [4, 6];*Positive* label to the cluster of images with valence values predominantly in the range [6, 9].

Once these three clusters were identified, specific pools of images had to be extracted in order to associate them with the seven emotional states. To do so, the discrete information in the database, namely the discrete labels of each basic emotion and the mean intensity values associated with each image, was utilized.

Firstly, to locate in which of the three clusters to search for the specific images for each discrete label, it was necessary to establish a relationship between each discrete emotion and the clusters previously created. Several methodologies for remapping discrete emotions in the dimensional space are reported in literature. The philosopher Russell J.A., known for developing the circumplex model of emotions, provided a characterization of discrete emotions in dimensional terms [40,41].

Moreover, statistical analysis methodologies [42,43] were employed to establish a connection between the two ways of interpreting emotions, such as associating discrete emotion labels with corresponding valence and arousal value pairs. For instance, the emotion label “happiness” may be linked to a specific combination of valence and arousal values. We combined various studies on literature to evaluate how the discrete emotions were distributed in the dimensional space and identify one or more clusters to search for images for each specific discrete emotion, as shown in Table 1.

Considering Table 1, a noteworthy distinction arises in the treatment of images associated with the emotion of ‘surprise’ compared to those linked to other emotional states. Specifically, whereas images correlated with other emotional states were confined to particular clusters, images linked to the ‘surprise’ emotion were deliberately examined across all clusters. This strategic decision is rooted in the inherent universality of the ‘surprise’ emotion. Research examining the dualistic nature of discrete emotions [44,45], underscores the necessity of an accurate conceptualization of the ‘surprise’ emotion that embraces both its positive and negative components. In contrast to the categorical classifications typical of other emotions, the complexity of ‘surprise’ arises from the simultaneous interweaving of these opposing dimensions. This comprehensive perspective elucidates the rationale underlying the distinctive approach adopted when seeking images associated with the ‘surprise’ emotion, as illustrated in Table 1.

The images associated with each specific emotion were searched in the corresponding clusters, considering only those elements that the database developers had labeled with at least the same discrete label as the emotion under consideration. For each emotion, the isolated elements were then rearranged in descending order according to the intensity associated with the emotion in question, in order to select the first ten images to be used for composing the database used in the test phase.

However, it should be noted that:Before searching the images for each emotion from the specific clusters, a pre-screening procedure was carried out on the images. In fact, to avoid excessively shocking the subjects’ sensitivity, images that had been assigned an intensity value associated with the emotion “disgust” greater than 4 by the database developers were not considered.The neutral emotional state is the only one for which it was not possible to isolate the ten images considering only the labels and intensity values, as the latter information is not provided in the database information table. Therefore, an alternative procedure was implemented:It was decided to use the discrete labels and arousal values provided.Images belonging to the neutral cluster, for which the database developers had assigned all six basic emotion labels, were considered.These images were then reordered according to arousal values, using an increasing sorting order.Finally, the top ten images with the lowest arousal values were isolated.

In the end, the application of this procedure allowed us to identify ten images for each of the discrete emotions exploited by the face-api.js library (i.e., neutrality, happiness, surprise, fear, sadness, anger, and disgust), resulting in a total of 70 images to be used in the experimental protocol.

### 2.3. Web Application

The test was administered through a web app, where both the front-end and back-end were developed using JavaScript. Specifically, the front-end was developed using p5.js, while the back-end was developed using the JavaScript runtime environment Node.js. Consequently, the data obtained throughout the testing phase was saved in the cloud database hosted by the Google Firebase platform.

The decision to utilize p5.js, an open-source Javascript library that provides a comprehensive set of graphical tools, allowed for the creation of a visually engaging and interactive interface. This enhanced the overall user experience and facilitated the seamless execution of the research. In fact, before administering the test to the subjects, an evaluation of the design and acceptability of the developed application was conducted to validate its efficacy. In this validation test, laboratory students participated, and they provided positive feedback on the interface, describing it as user-friendly, intuitive, and visually appealing.

Specifically, the interface comprises a sequence of user-friendly pages, ensuring inclusivity, explanation, and ease of use throughout the clinical trial experience. The first page provides a brief summary of the activity to be carried out and an input section where the assigned users’ identification code is entered, followed by a specific button to start the test. Once the test is started, the emotional image is displayed full screen for 3 s. The interface was developed in order to enable the FER algorithm under investigation (i.e., face-api.js) during this time interval. In this way, it is possible to detect the emotional states expressed through facial expressions for each image and the crucial points of the face for each subject. After this interval, the page for selecting emotional states appears automatically. This page features a smaller version of the image and seven buttons that correspond to the seven considered emotions, allowing the user to select the emotions, one or more, experienced while viewing the image. The transition to the next image occurs without time constraints through a specific button, enabled only after the user has selected at least one emotional state. A 3 s break page has been added before viewing the next image to update participants on the progress of the test. Upon the expiration of this time, automatic redirection to the next page takes place. The choice to set these intervals to 3 s was made to ensure contextual consistency with the experimental protocol adopted by the developers of the highly emotive image content database used in the study [37].

The same sequence of procedures described for the first image is repeated until all 35 images have been viewed and evaluated. The test concludes with a final page that informs the user of the test’s completion and expresses gratitude for their participation.

In conclusion, based on the feedback provided by the test subjects, it can be asserted that the interface is easily understandable, intuitive, and visually appealing. Figure 2 shows some of the pages of the user interface just described.

### 2.4. FeelPix Dataset

During the validation process, specific participant information was extracted and utilized for the development of a labeled dataset. For the 31 healthy participants (13 females and 18 males) ranging in age from 20 to 69, as depicted in the distribution chart in Figure 1, two-dimensional coordinates of the 68 facial landmarks were extracted. It was chosen to isolate the information associated with landmarks as they represent reference points that allow for identifying and tracking particular facial features, including facial expressions [46,47]. Additionally, emotional states, specified by each participant for every presented image, were collected. These data served as the foundation for constructing the labeled dataset.

This allowed us to have, for each dataset sample, a set of 68 x,y coordinates, accompanied by labeling determined by the ground truth. The ground truth was derived from the emotions specified by users during the experimental phase, resulting in seven labels that characterize each dataset sample. Each of the seven ground truth labels corresponds to one of the seven emotions that the user could indicate in response to the emotional state induced by viewing the image. Binary classification was utilized to translate user selections into discrete data, where a value of 1 was assigned to the chosen emotion labels, while all unselected emotions were labeled with a value of 0. This procedure was applied to each sample, ensuring a consistent mapping between sets of coordinates and a sequence of 1 s and 0 s corresponding to the sample’s labels, resulting in a meticulously labeled dataset.

Once the ground truth for the dataset was constructed, to make the raw data detected compatible with the most common standards, a processing procedure was implemented. As a first step, all landmark coordinates were normalized to reduce the impact of different subjects’ face positions in the webcam’s field of view and their different physiognomy. To achieve this, four specific anatomical points of the face, whose position is not influenced by facial expression, were identified:*Left meningeal*;*Right meningeal*;*Nasal center*;*Subnasal center*.

The position of these landmarks is highlighted with four red circles in Figure 3a, wherein the coordinates are reported in the plane prior to their normalization.

So, the distances between pairs of these points were used to normalize the coordinates of all 68 landmarks. Specifically, the horizontal distance between the left meningeal point and the nasal center point was used to normalize the *x* coordinates of all points to the left of the nasal center point. Similarly, the *x* coordinates of all points to the right of the nasal center point were normalized using the horizontal distance between this point and the right meningeal point. The use of two separate distances for the horizontal normalization of the points on the left and right of the nasal center point allowed to balance the symmetry between the right and left portions of the face. The vertical distance between the nasal center point and the sub-nasal center point was used to normalize the *y* coordinates of all points, regardless of their position.

After applying coordinate normalization, a selection procedure was implemented to identify a reduced number of points deemed to be highly informative. Among all existing AUs, the FACS manual [11] isolates 22 considered fundamental as they describe specific facial muscles responsible for human facial expressions. In particular, the importance of these 22 facial muscles for understanding human expressions is highlighted as they are responsible for the contraction and relaxation of different parts of the face, thus their movement is closely related to facial expressions. For these reasons, only the coordinates of the 22 points related to these specific muscles, and their movement, were considered for the realization of the database. This allowed for the isolation of only the most significant points for the considered applications, resulting in a highly performant database optimized for its specific purpose. The aforementioned steps can be observed in the sequence of images presented in Figure 3.

The described procedure was implemented following a meticulous performance evaluation, which involved employing classification algorithms to assess the dataset’s efficiency, as detailed in the following section. The evaluation conducted revealed the presence of samples with limited informative content regarding emotional expressiveness. More precisely, including such samples introduced significant bias, compromising performance. Consequently, we decided to reduce the number of features associated with each sample, considering only 22 landmarks. This reduction not only improved performance but also alleviated computational load, enabling faster analysis.

This database, named *FeelPix*, is available on GitHub.com [48] for other researchers and developers to use for the development of facial expression recognition algorithms based on landmarks.

Compared to existing datasets for facial expression recognition, which primarily consist of images, our dataset provides detailed data based on facial landmarks, eliminating the need for further processing through Convolutional Neural Networks or facial landmark coordinate extraction. Moreover, while most existing datasets have labels generated by developers, FeelPix stands out for collecting the data labels directly from the participants involved in the study, minimizing the potential errors or biases introduced by third-party labeling that did not personally experience the captured emotional states during detection.

### 2.5. Testing Algorithm

To assess the effectiveness of the developed dataset, a simple Facial Expression Recognition algorithm was created and trained using this data. The dataset was processed to reduce the classification problem to a binary one, where:The selection of a specific emotion was associated with class 1;The absence of the specific emotion’s selection was associated with class 0.

In this manner, an algorithm employing seven machine learning classifiers, each optimized for a specific emotion, was developed. The classifiers utilized in this study include Support Vector Machine (SVM) and Random Forest (RF), chosen for their efficiency and lower data requirements compared to neural networks.

However, it emerged that the number of samples between the two classes was highly imbalanced, with a larger quantity of samples in class 0 compared to class 1. This imbalance is attributed to class 1 representing the selection of the specific considered emotion, while class 0 encompasses the non-selection of that specific emotion but the possibility of selecting all other emotions. Consequently, before training the classifiers, a final dataset processing step was performed to mitigate the extreme imbalance. The undersampling approach was adopted, keeping all data in the minority class and reducing the size of the majority class.

This approach was employed by considering the number of positive samples in the dataset under examination and randomly selecting a specific number of negative samples, thus facilitating the construction of a less imbalanced dataset. In particular, since the negative samples represent a greater variety of emotions than the positive samples, a partial approach was adopted: the difference in samples between the two classes was appropriately regulated and set to half the number of samples present in the minority class.

Subsequently, an optimization of both considered classification methodologies was carried out for each emotion, through a random search of hyperparameters. Lastly, both methodologies were trained on the developed database, to determine which of the two ones yielded the best performance for each emotion.

To achieve this, a five-fold cross-validation was conducted, considering accuracy, precision, F-measure, and G-mean as validation metrics. Precision, F-measure, and G-mean were chosen because they are insensitive to the dataset’s imbalance, which, although mitigated through undersampling, still exhibited slight imbalances.

## 3. Results

In our study, we had three primary objectives. Firstly, we aimed to develop a universal methodology for evaluating the performance of Facial Expression Recognition algorithms. Subsequently, we collected labeled data during tests to create a comprehensive dataset. Finally, we focused on developing an FER algorithm capable of verifying the reliability of the created database by classifying facial expressions based on facial landmark coordinates.

The results will demonstrate the achievement of these objectives by showcasing the algorithm’s performance using the developed validation protocol. Additionally, the validity of the released dataset and the proposed algorithm will be highlighted by presenting the algorithm’s performance on the dataset’s data. Furthermore, a comparison between the two performances will be provided.

### 3.1. Investigated Algorithm Results

To assess the reliability of the investigated FER algorithm, i.e., the one implemented in the face-api.js library, the success rate for each user was initially calculated by comparing their selected emotions with the library’s detections for each image. Specifically, this quantity was determined as the ratio between the number of images in which at least one of the user’s choices and at least one of the algorithm’s detections agreed with each other, and the total number of images displayed to the subject (i.e., 35 images).

Unexpectedly, the results obtained were highly variable among different users and, on average, below 54%. Given the significant deviation from the expected outcome, it was deemed appropriate to calculate the same validation metrics used to evaluate the testing algorithm’s performance for each emotion.

However, as reported in Table 2, even these values were considerably lower than expected: the metrics assume percentages below 40% in most cases and never exceed 60%, as shown in Figure 4.

### 3.2. Outcomes of the Dataset Testing

The FeelPix dataset was utilized to train and test the algorithm developed for its evaluation. This enabled determination of the performance achievable using such data. In order to conduct a comprehensive investigation, validation metrics were computed for both considered classification methodologies (i.e., SVM and RF).

This facilitated identification of the methodology exhibiting the best performance for each emotion, thereby composing the algorithm that recognizes all seven emotions from the database data. The validation outcomes of the various classifiers on the created database dataset met the expectations, as the metrics exhibit values higher than 75% in most cases and never lower than 64%, as evident from Table 3 and Figure 5, where the values associated with the best performing classifier for each emotion are presented.

### 3.3. FER Algorithms’ Comparison

It is evident that the validation metric values obtained using the combination of the developed database and proposed testing algorithm are significantly better than the results shown by the FER algorithm under investigation, as depicted in the Figure 6.

Conclusively, the developed and released database proves to be efficient and functional, as it achieves sufficiently high performance even with a simple classification algorithm.

## 4. Discussion

In this work, we have investigated the world of emotions, specifically their recognition through facial expression analysis. In light of the current state-of-the-art in emotion recognition, it has emerged that detecting emotions in naturalistic conditions still presents significant difficulties. Numerous variables contribute to these difficulties, such as the lack of a universally accepted definition of facial codes and/or facial actions. Furthermore, the comprehensive understanding and prediction of affective processes undoubtedly requires careful integration of multiple contextual factors, information modalities, and evaluations within naturalistic environments. Therefore, evaluating and verifying FER algorithms is a challenging but critical task.

For this reason, we developed a protocol to validate FER algorithms in this work. In the protocol, participants were shown static emotional stimuli on a computer screen and asked to select the emotion they felt while the facial expression was detected by the FER algorithm. The test was performed through a web application, which proved to be functional, easy to use, intuitive, and visually appealing. Furthermore, the application did not encounter any problems during data acquisition, and the results were correctly saved for all subjects in the selected database. Therefore, the developed system can be considered an effective tool for validating FER libraries. Furthermore, the proposed technology also enables the creation of a database of spontaneous expressions that incorporates various contextual factors, such as different genders, ethnicities, personalities, and cultures.

The validation of the FER algorithm implemented in the JavaScript library face-api.js used in this study yielded results that were not in line with the initial expectations. In fact, the algorithm’s performance was significantly inferior to the generally accepted values, indicating a poor ability to correctly recognize facial expressions. However, it is important to consider that these results are limited by the context of the type of stimulus and the experimental setup. Additionally, it is worth noting that the results obtained may also be influenced by the selection of participants and their degree of expressiveness.

On the other hand, the labelled data database, created by processing the landmark coordinates provided by the algorithm under examination and the user choices, proved to be accurate, functional, and reliable, as demonstrated by the validation metrics achieved by the testing algorithm applied on it. In fact, the combination of the proposed testing algorithm and the developed database allows for the recognition of all seven emotions with high accuracy, making it a valuable tool in the field of behavioral sciences and facial expression recognition.

It is important to note that the data used to create the database were collected under experimental conditions where a constant level of expressiveness of the subjects was not guaranteed, thereby generating a dataset that covers a wide range of conditions. Therefore, considering that this database has successfully captured even the most subtle facial expressions, it can be deemed as a valuable asset for a range of potential applications, such as human-computer interaction, affective computing, and mental health diagnosis. Furthermore, in future developments, minor changes, such as expanding the type of stimulus or the experimental environment, would allow more information to be added to the database, thus enabling the proposal of a tool that can integrate the multiple contextual factors that typically make facial expression recognition challenging.

## 5. Conclusions

In conclusion, we have presented a comprehensive validation system for Facial Expression Recognition (FER) algorithms, offering several significant advantages. Firstly, the system’s applicability to any type of FER algorithm allows for the efficient determination of its effectiveness, reducing the algorithm verification phase, and ensuring the utmost accuracy and reliability in identifying and interpreting facial expressions. Moreover, the system provides a clear and transparent description of the algorithm’s proficiency in recognizing and interpreting facial expressions, facilitating the identification of any potential bias or errors. Additionally, the validation system supports the development of new solutions and applications, fostering a deeper understanding of its capabilities and limitations, which is invaluable for advancing the field of FER.

Furthermore, as part of our contributions, we have developed and released a meticulously labelled data database, which bestows various advantages. The database offers extensive coverage of a wide range of conditions, encompassing both highly expressive and less expressive subjects, providing a rich and diverse dataset for research and experimentation. Accessible and user-friendly, the database empowers researchers and professionals to readily employ it in their studies and application development pertaining to emotion recognition through facial expression analysis.

Finally, the algorithm we have devised for database verification enables the precise identification and interpretation of emotions associated with facial expressions, offering profound insights into individuals’ emotional reactions. Consequently, it proves instrumental in facilitating more natural and intuitive interactions between humans and technological interfaces, such as robots, virtual assistants, and augmented reality systems, making it a powerful tool for psychological and social research.

By combining these contributions, our work advances the frontiers of facial expression recognition, paving the way for enhanced emotional understanding and human-computer interactions in various domains.

## Figures and Tables

**Figure 1 sensors-23-08376-f001:**
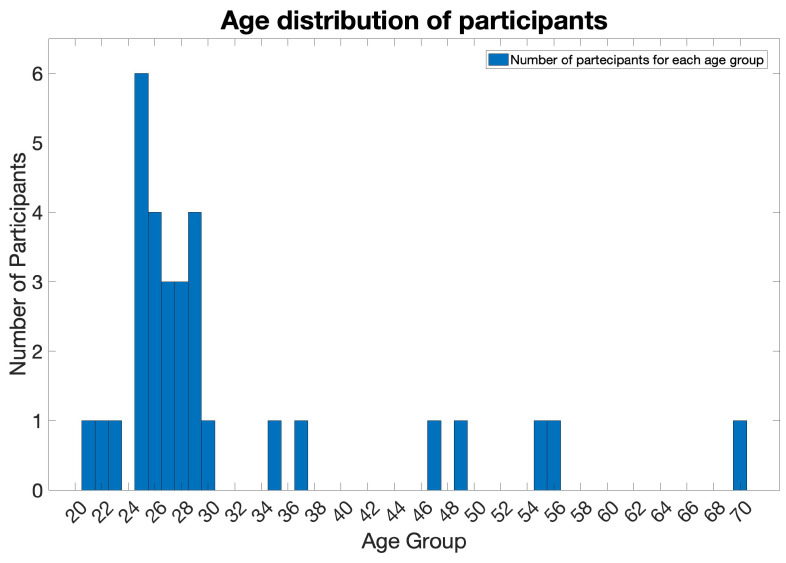
Age distribution among individuals involved in the experimental validation process of the Facial Expression Recognition algorithm under investigation.

**Figure 2 sensors-23-08376-f002:**
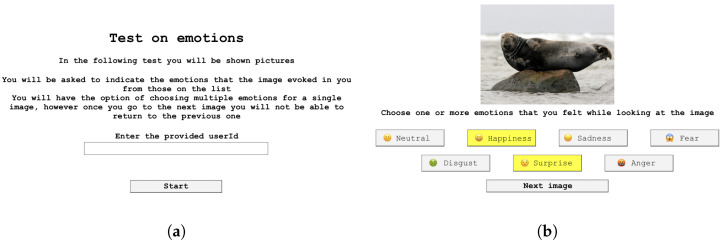
Examples of the graphical interface pages designed for the experimental protocol: (**a**) example of the starting page; (**b**) example of the selection page. (Translated from the original Italian version).

**Figure 3 sensors-23-08376-f003:**
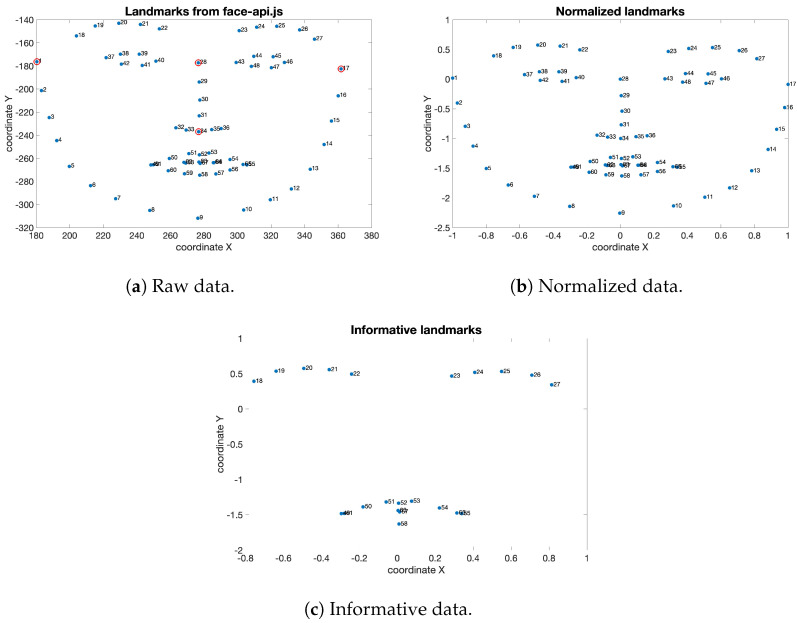
Facial landmarks coordinates elaboration process: (**a**) coordinates of the landmarks as obtained by the algorithm under investigation; (**b**) coordinates after applying normalization; (**c**) coordinates of the 22 key points selected for their high degree of informativeness.

**Figure 4 sensors-23-08376-f004:**
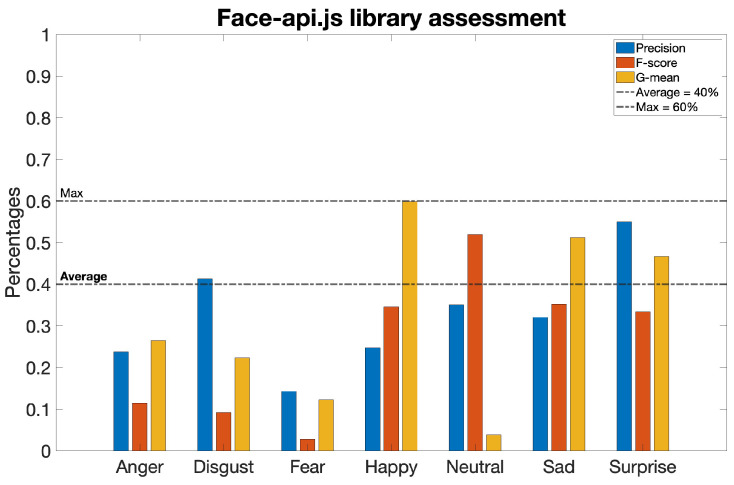
Validation metrics—Precision, F-score, and G-mean—pertaining to the investigated algorithm during the validation process. These metrics are presented for each discrete emotion category, including neutrality.

**Figure 5 sensors-23-08376-f005:**
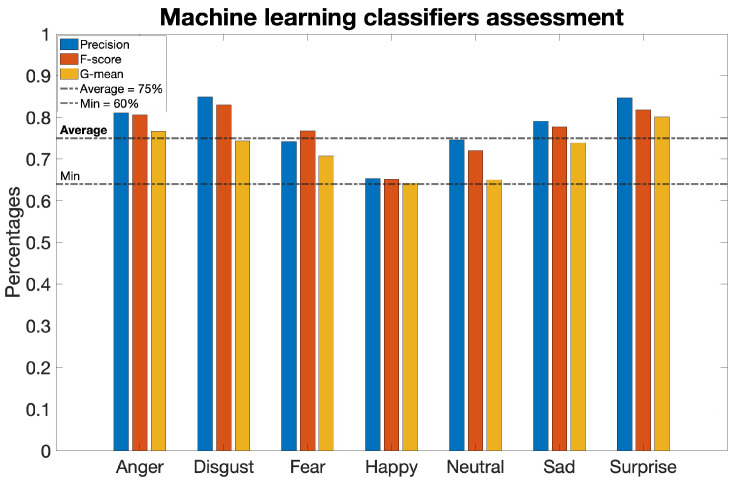
Validation metrics acquired from the algorithm developed to ascertain the integrity of the proposed FeelPix database. These metrics include precision, F-score, and G-mean, and are displayed for each emotion category.

**Figure 6 sensors-23-08376-f006:**
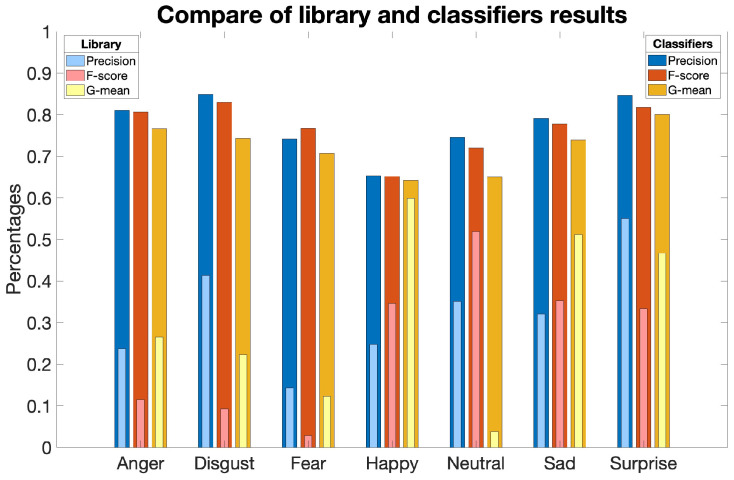
Comparing the outcomes produced by the algorithm under investigation during the validation process with those achieved by the algorithm developed for validating the proposed database. The visualization includes precision, F-score, and G-mean metrics, shedding light on the performance of each algorithm across diverse emotion categories.

**Table 1 sensors-23-08376-t001:** Results of the mapping between discrete and dimensional emotions, with cluster identification for each label.

Discrete Emotions	Valence [1, 9]	Arousal [1, 9]	Cluster
Anger	3.2	7.7	*Negative*
Disgust	2.6	6.4	*Negative*
Fear	2.4	7.4	*Negative*
Happy	8.0	6.9	*Positive*
Neutral	5.0	3.0	*Neutral*
Sad	2.4	6.1	*Negative*
Surprise	6.6	7.7	All clusters

**Table 2 sensors-23-08376-t002:** Validation metrics for the algorithm under examination.

	Anger	Disgust	Fear	Happy	Neutral	Sad	Surprise
Precision	24%	41%	14%	25%	35%	32%	55%
F score	11%	9%	3%	35%	52%	35%	33%
G mean	26%	22%	12%	60%	4%	51%	47%

**Table 3 sensors-23-08376-t003:** Validation metrics for the testing algorithm.

	Anger	Disgust	Fear	Happy	Neutral	Sad	Surprise
Accuracy	78%	78%	72%	65%	67%	74%	80%
Precision	81%	85%	74%	65%	75%	79%	85%
F score	81%	83%	77%	65%	72%	78%	82%
G mean	77%	74%	71%	64%	65%	74%	80%

## Data Availability

The dataset developed in this study is fully available as an open-source resource. Interested parties can access the complete dataset on the GitHub platform, link https://github.com/ludovicalamonica/FeelPix (accessed on 18 April 2023).

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
