# Peer review of "Development of a Universal Validation Protocol and an Open-Source Database for Multi-Contextual Facial Expression Recognition"

_sensors, 2023, doi:10.3390/s23208376_

Round 1
Reviewer 1 Report
The manuscript presents a comprehensive study on the development of a universal validation methodology for Facial Expression Recognition (FER) algorithms. The authors have made a commendable effort in introducing a web application-based approach and the creation of the FeelPix database. The introduction and related work sections provide a solid foundation for the reader to understand the context and significance of the research.
The topic is timely and relevant, especially in the age of AI and human-computer interaction.
The authors have provided a thorough review of existing FER algorithms, emotion theories, and technologies, which helps in understanding the current state of the art.
The objectives of the study are well-defined and set a clear direction for the research.
While the materials and methods section provides an overview of the experimental setup, it would benefit from a more detailed explanation of the selection criteria for the emotive images and the rationale behind choosing a subset of 35 images for each participant.
The introduction of the FeelPix database is promising. However, more details about the database, such as the number of participants, diversity in age, gender, ethnicity, and how the data was labeled, would provide clarity.
The manuscript mentions the use of the face-api.js library. A deeper dive into its technical aspects, advantages over other libraries, and its limitations would be beneficial for readers unfamiliar with this tool.
Overall, the manuscript offers valuable contributions to the field of Facial Expression Recognition. With some elaboration in the mentioned areas, it has the potential to be a significant reference for researchers and professionals in the domain.
Author Response
Thank you for your valuable comments. Below, we provide point-by-point responses.
The manuscript presents a comprehensive study on the development of a universal validation methodology for Facial Expression Recognition (FER) algorithms. The authors have made a commendable effort in introducing a web application-based approach and the creation of the FeelPix database. The introduction and related work sections provide a solid foundation for the reader to understand the context and significance of the research.
The topic is timely and relevant, especially in the age of AI and human-computer interaction.
The authors have provided a thorough review of existing FER algorithms, emotion theories, and technologies, which helps in understanding the current state of the art.
The objectives of the study are well-defined and set a clear direction for the research.
While the materials and methods section provides an overview of the experimental setup, it would benefit from a more detailed explanation of the selection criteria for the emotive images and the rationale behind choosing a subset of 35 images for each participant.
We agree with the reviewer's assessment and recognize the need to provide a rationale for presenting participants with a set of 35 images. This decision was based on a thorough assessment of the test's timing. During the study's design phase, we conducted pilot tests to determine the average time required for a participant to complete the emotional test using various quantities of images while avoiding potential fatigue. As a result, we have revised the specific section to emphasize these motivations (lines 201 – 209).
The introduction of the FeelPix database is promising. However, more details about the database, such as the number of participants, diversity in age, gender, ethnicity, and how the data was labeled, would provide clarity.
Based on the provided feedback, we have made the necessary revisions to enhance the introduction of the FeelPix database (lines 420 – 437). Specifically, we have incorporated additional details regarding the participants involved in data collection. Additionally, we have provided a more detailed explanation of how data labeling was conducted, emphasizing the process that led to the creation of ground truth based on emotions expressed by the participants. We believe that these modifications will improve the understanding of the proposed dataset.
The manuscript mentions the use of the face-api.js library. A deeper dive into its technical aspects, advantages over other libraries, and its limitations would be beneficial for readers unfamiliar with this tool.
We understand the reviewer's comment and have made the necessary updates in our article to provide more details about the face-api.js algorithm (lines 91 – 102). Given the limited information available regarding this algorithm, we share the importance of including this information to assist readers who may not be familiar with this tool. Face-api.js was selected for testing precisely because, despite its use in various research studies, it lacks comprehensive evaluation documentation.
Overall, the manuscript offers valuable contributions to the field of Facial Expression Recognition. With some elaboration in the mentioned areas, it has the potential to be a significant reference for researchers and professionals in the domain.
Reviewer 2 Report
The first work of this paper is to develop a web-based validation method to evaluate the performance of the facial expression recognition (FER) algorithm. The second work is to collect the facial expression data of the subjects in the test phase, so as to make a facial expression dataset, called FeelPix. The third work is to develop a FER algorithm by improving the 68 facial landmarks method. Here are the reviews for this article:
1. This paper can further discuss the specific differences between the proposed dataset and the existing dataset.
2. This paper should give a clearer description of the three goals mentioned in lines 122-153 in Section 3 Result, and how to achieve these three goals.
3. The first goal of this article is "For this reason, the first goal of this work is to devise a universal validation methodology, applicable to any algorithm, aimed at evaluating the performance of Facial Expression Recognition (FER) algorithms. "This article is only for face-api.js library is an algorithm for comparison. Therefore, other algorithms should be added to verify the effectiveness of the proposed method.
4. The developed algorithm is based on the 68 facial landmarks method, which uses normalization and selection of key feature information points, and then complete classification by SVM or RF. However, this paper does not point out whether the advantages of the developed algorithm compared with the 68 facial landmark algorithm have any impact on the accuracy and influence speed and other indicators.
5. Some parts of the page layout are unreasonable, and large areas cannot be left blank, such as P7 and P12.
6. The cited literature is not novel enough, and some work from the past two years can be reintroduced.
Well written-presentation.
Author Response
Thank you for your valuable comments. Below, we provide point-by-point responses.
The first work of this paper is to develop a web-based validation method to evaluate the performance of the facial expression recognition (FER) algorithm. The second work is to collect the facial expression data of the subjects in the test phase, so as to make a facial expression dataset, called FeelPix. The third work is to develop a FER algorithm by improving the 68 facial landmarks method. Here are the reviews for this article:
- This paper can further discuss the specific differences between the proposed dataset and the existing dataset.
In accordance with the reviewer's request, in the section dedicated to dataset explanation, we have specified the distinctive advantages of our dataset compared to existing ones (lines 483 – 489). In particular, we believe that the proposed dataset eliminates the need for additional data preprocessing before being utilized in algorithms for facial expression recognition based on facial landmarks. Furthermore, the data were collected and labeled using information provided by the same subjects who participated in the study, thus reducing potential errors and biases stemming from external labeling.
- This paper should give a clearer description of the three goals mentioned in lines 122-153 in Section 3 Result, and how to achieve these three goals.
As recommended by the reviewer, we have now provided a description of the three objectives and how achieve them within the Results section (lines 522 – 531). We agree with the reviewer that this will enable a better understanding of the presented data and how these objectives were achieved.
- The first goal of this article is "For this reason, the first goal of this work is to devise a universal validation methodology, applicable to any algorithm, aimed at evaluating the performance of Facial Expression Recognition (FER) algorithms. "This article is only for face-api.js library is an algorithm for comparison. Therefore, other algorithms should be added to verify the effectiveness of the proposed method.
We express our gratitude to the reviewer for his valuable feedback. While our article primarily focuses on validating the algorithm implemented within the face-api.js library, we acknowledge the importance of testing alternative facial expression recognition (FER) algorithms to comprehensively evaluate the proposed methodology. However, we need to note that we cannot immediately incorporate additional results as they require new user experiments, which could be implemented for a successive work. Nevertheless, we want to emphasize that the inclusion of the evaluation of new FER algorithms in our research is a goal we intend to pursue. This commitment will facilitate the enhancement of both the validity and applicability of the proposed validation methodology. Consequently, we are dedicated to a long-term effort aimed at enriching our research.
- The developed algorithm is based on the 68 facial landmarks method, which uses normalization and selection of key feature information points, and then complete classification by SVM or RF. However, this paper does not point out whether the advantages of the developed algorithm compared with the 68 facial landmark algorithm have any impact on the accuracy and influence speed and other indicators.
In line with your feedback, we have now included in the paper the advantages we observed using a reduced number of landmarks (lines 472 – 479). Specifically, our decision to utilize only 22 landmarks, as opposed to the conventional 68 used by some existing facial expression recognition algorithms, is justified by the improvements in terms of efficiency, reliability, and computational performance achieved by our algorithm. These enhancements contribute positively to accuracy and processing speed, which are key indicators for our applications.
- Some parts of the page layout are unreasonable, and large areas cannot be left blank, such as P7 and P12.
The page layout has been thoroughly reviewed and refined to address concerns about the excessive blank areas, such as in P7 and P12, resulting in a more balanced and reasonable presentation, as suggested by the reviewer.
- The cited literature is not novel enough, and some work from the past two years can be reintroduced.
Thank you for bringing this gap to our attention, and we appreciate your suggestion. We have taken the necessary steps to update the references in our work, incorporating more recent research.